# Knowledge Accumulating Contrastive Prompt for Continual Learning

## Abstract

Continual learning has been challenged by the issue of catastrophic forgetting (CF). Prompt-based methods have recently emerged as a promising approach to alleviate this problem, capturing the previous knowledge by the group of prompts. However, selecting an appropriate prompt during the inference stage can be challenging, and may limit the overall performance by the misaligned prompts. In this paper, we propose a novel approach to prompt-based continual learning, which accumulates the knowledge in a single prompt, which has not been explored previously. Specifically, inspired by contrastive learning, we treat the input with the current and previous prompt as two different augmented views (i.e., positive pair). We then pull the features of the positive pairs in the embedding space to accumulate knowledge. Our experimental results demonstrate the state-of-the-art performance in continual learning even with a single prompt, highlighting the potential of this approach towards a 'holistic' for the model.

## 1 Introduction

The primary objective of continual learning, also known as lifelong learning (Silver & Mercer, 2002; Rannen et al., 2017), is to learn the complete knowledge for a set of tasks when each task is presented sequentially. This is particularly relevant in real-world applications where the tasks to be adapted change over time. However, neural networks tend to forget the previously acquired knowledge as the model parameters are optimized for the new tasks, leading to the problem of catastrophic forgetting. To address this challenge, many researchers have focused on developing approaches to alleviate the forgetting problem of the neural network during the continual learning. Previously, three main approaches have been suggested. Regularization-based approaches restrict parameter updates according to their relevance to previous tasks (Kirkpatrick et al., 2017; Zenke et al., 2017; Lee et al., 2017; Li & Hoiem, 2017; Buzzega et al., 2020). Rehearsal-based approaches repeatedly use previous task data during training for a new task, either through memory buffers or by generating the previous data (Lopez-Paz & Ranzato, 2017; Chaudhry et al., 2019a; Saha et al., 2021; Prabhu et al., 2020). Architecture-based approaches increase model capacity by adding new modules for new tasks (Yoon et al., 2018; Yan et al., 2021). The rehearsal-based methods outperforms the other approaches, however, the methods are not free from the data privacy issue as the training procedure entails the exposure of the previous data.

Recently, the prompt-based methods for the continual learning have gained attention due to its superior performance, without any concerns on the data privacy issue. The prompt is an additional input texts or images, which adapts the model for specific tasks by the small number of parameters. Inspired by the promising ablity of the prompt for the model adaptation, many researchers have proposed the method to capture the previous knowledge by the prompt for the continual learning (Wang et al., 2022c;b). Specifically, the methods utilize the prompt pool which stores the prompts optimized by the previous tasks. During inference, a prompt is selected from the pool to instruct the model, providing the previous knowledge. The prompt-based methods presents the effective way to address the forgetting of the pretrained-large scaled model by the small number of learnable parameters. Moreover, the methods are free from the data privacy issue, which is a limitation of the rehearsal-based approach. However, the prediction for the task identity is required to select the prompt for the inference, which can introduce misaligned prompts and limit the performance.

Therefore, in this paper, we present a method for continually accumulating knowledge into a single prompt, which does not require a prompt pool or the task identity prediction. Our work is motivated by the existence of the upper-bound prompt for the sequence of the tasks, as illustrated in Figure 1. Specifically, when all the tasks are presented simultaneously, the prompt $p^*$ is optimized to instruct the model to adapt for all the tasks. As the existence of the $p^*$ is ensured, the primary concern is to approach the $p^*$ effectively when the tasks are given sequentially in the continual learning.

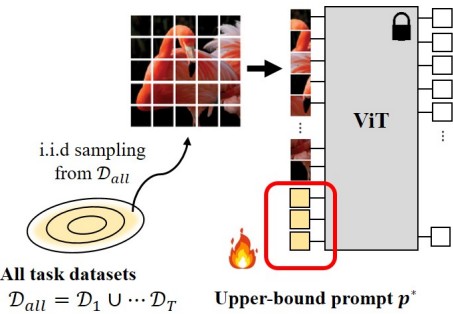

Figure 1: When all tasks are given simultaneously, the obtained prompt $p^*$ guides the model to adapt for all tasks.

Accordingly, we propose the knowledge accumulating contrastive prompt which is the novel approach for the prompt-based continual learning. Specifically, we propose to utilize the current and previous prompts as two augmented views for the input (i.e. positive pair) to increase the dependency between them. Inspired by contrastive learning without negative pairs (Chen & He, 2021; Grill et al., 2020), we use a similarity-based loss to pull the positive pairs closer in the embedding space. We first investigate how the model's features change by the prompt during continual learning, and how this leads to forgetting of previous knowledge. Then, we propose our method to accumulate knowledge in a single prompt by increasing the dependency between the current and previous prompts, which helps to alleviate the forgetting problem.

Our contribution can be summarized as follows:

- We propose a novel approach for prompt-based continual learning, progressively approaching the upper-bound prompt by the sequence of the tasks.

- We present a methodology for accumulating task knowledge in a single prompt, eliminating the need for a prompt pool or task prediction during inference.

- We suggest a method that leverages the contrastive learning between the current and previous prompts, constructing a positive pair by the prompts.

- We demonstrate state-of-the-art results using the single prompt, highlighting the potential of our approach to approach the upper-bound prompt for the sequence of the tasks.

## 2 RELATED WORKS

**Continual learning** Continual learning aims to enable a network to adapt over time to a sequence of tasks without forgetting previously acquired knowledge. However, the network often forgets previously learned knowledge as it is trained to perform well on new tasks, regardless of the degradation of its previous performance. This problem is known as the *catastrophic forgetting* problem, which is a major challenge in continual learning. The causes of the forgetting problem are attributed to the changes in important parameters (Kirkpatrick et al., 2017; Zenke et al., 2017), interference by gradient direction (Lopez-Paz & Ranzato, 2017; Saha et al., 2021), or the change in the feature space (Zhu et al., 2021). To alleviate the forgetting problem, many methods have been proposed and can be categorized into three approaches. Firstly, regularization-based methods (Kirkpatrick et al., 2017; Zenke et al., 2017; Lee et al., 2017; Buzzega et al., 2020) enforce the model to preserve previously learned knowledge through additional regularization techniques. For example, EWC (Kirkpatrick et al., 2017) and SI (Zenke et al., 2017) regularize the change of model parameters based on their importance, while IMM (Lee et al., 2017) utilizes moment matching. Knowledge distillation loss is also used to preserve previous knowledge (Li & Hoiem, 2017; Buzzega et al., 2020). Secondly, rehearsal-based methods are presented, which repeatedly train the model using previous task data. For instance, the replay buffer is utilized to memorize previous task data to compute gradients for the previous task (Lopez-Paz & Ranzato, 2017; Chaudhry et al., 2019a) or previous eigenvectors (Saha et al., 2021). Generative approaches are suggested to replay the generated previous task data (Shin et al., 2017; Rao et al., 2019) to alleviate the forgetting problem. Thirdly, architecture-based methods are suggested to dynamically increase the model's capacity as new tasks arrive (Yan et al., 2021;

Yoon et al., 2018). Moreover, some recent works (Cha et al., 2021; Madaan et al., 2022) suggest that unsupervised learning can learn unforgettable representations that may help alleviate the forgetting problem. Lastly, the prompt-based methods (Wang et al., 2022b;c) are suggested to capture task-specific knowledge by using a prompt pool to memorize previous task knowledge.

**Contrastive learning**    Contrastive learning methods have proven effective in capturing useful representations by utilizing positive and negative pairs. The widely used InfoNCE loss (Oord et al., 2018; Tian et al., 2019) maximizes the mutual information between pairs by pulling and pushing positive and negative pairs in the embedding space. However, InfoNCE loss requires a large number of negative pairs, which limits its effectiveness. As a result, positive-only methods that do not require negative pairs have been proposed (Grill et al., 2020; Chen & He, 2021; Bardes et al., 2022), which capture common features of positive pairs and increase their similarity. Some researchers have claimed that these methods capture a low-rank space that represents commonalities between pairs (Tian et al., 2021; Wang et al., 2021; Zhuo et al., 2023). More recently, researchers have suggested using contrastive learning to alleviate forgetting in continual learning (Guo et al., 2022; Madaan et al., 2022; Cha et al., 2021). Inspired by the recent approaches, the proposed method involves constructing prompt-augmented input pairs as positive pairs. Similar to positive-only methods, our approach pulls newly defined positive pairs to increase their commonalities and results in knowledge accumulation for the prompt.

## 3    MAIN CONTRIBUTION

### 3.1    MOTIVATION

In continual learning, a network is trained on a sequence of tasks $\mathcal{T}_{1:T} = \{\mathcal{T}_1, \ldots, \mathcal{T}_T\}$ with corresponding datasets $\mathcal{D}_{1:T} = \{\mathcal{D}_1, \ldots, \mathcal{D}_T\}$. Each task $\mathcal{T}_t$ has its own dataset $\mathcal{D}_t = \{(x_{i,t}, y_{i,t})\}_{i=1}^{n_t}$ consisting of $n_t$ pairs of input images $x_{i,t} \in \mathcal{X}_t$ and corresponding labels $y_{i,t} \in \mathcal{Y}_t$. As illustrated in Figure 1, our approach is motivated by the upper-bound prompt $p^*$ optimized by the merged dataset for all tasks (i.e. $\mathcal{D}_{all} = \mathcal{D}_1 \cup \cdots \cup \mathcal{D}_T$), as follows:

$$p^* = \arg\min_p \ \mathbb{E}_{(x,y) \in \mathcal{D}_{all}} \left[ L_{task}(f(x, p), y) \right] \tag{1}$$

where $L_{task}$ is the loss for the given task, which is the classification task in this paper. We point out that the $p^*$ is the solution for the continual learning, as the $p^*$ instructs the model to perform well for all tasks. Moreover, we argue that the $p^*$ refers the redundancy of parameters in previous approach based on the prompt pool, since the pool employs multiple prompts. Then, the primary objective centers on approaching the $p^*$ by the continual learning with a sequence of tasks $\mathcal{D}_{1:T}$. Accordingly, we propose the novel method that accumulates the task knowledge using the prompt, enabling a progressive approach towards $p^*$.

### 3.2    KEY OBSERVATION

We first investigate the process of catastrophic forgetting by prompt, analyzing the angle (Zhu et al., 2021) between the feature spaces of the current and previous models. Figure 2 illustrates the corresponding angle, with the task gap representing the difference between the current and previous task indices. Our results show that the top eigenvectors are forgotten during continual learning, as shown in Figure 2(b). This is consistent with the behavior of conventional training from scratch, as shown in Figure 2(a). Specifically, we observed that the top eigenvectors are transferred between consecutive tasks (i.e., when the task gap is 1), consistent with the prior work (Zhu et al., 2021). However, as the task gap increases, we found that the corresponding angle deviates, which is accompanied by

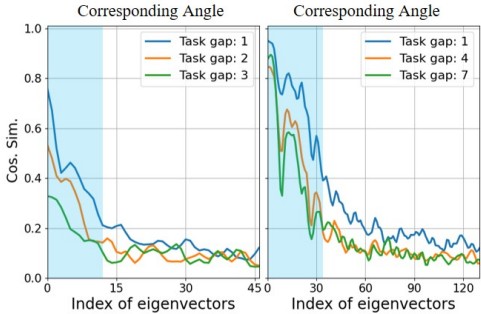

Figure 2: Colored regions indicate the forgetting of top eigenvectors. (a) ResNet model (b) prompt learning with ViT

a decline in task performance. Thus, we focus on aligning the feature space to preserve previous knowledge. The detailed settings for the observation is described in the Appendix A.

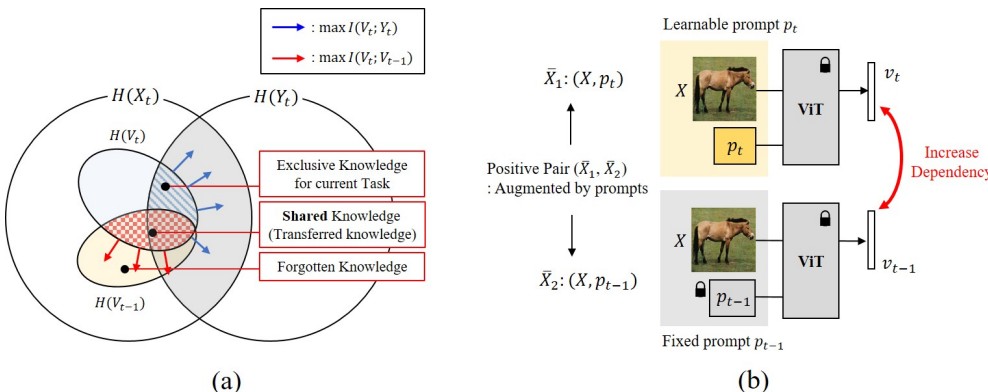

(a)                 (b)

Figure 3: (a) We maximize the shared knowledge $I(V_t, V_{t-1})$ using the prompt to alleviate the forgetting problem. (b) Proposed concept to progressively accumulate the task knowledge, by increasing the dependency between the current prompt $p_t$ and the previous prompt $p_{t-1}$.

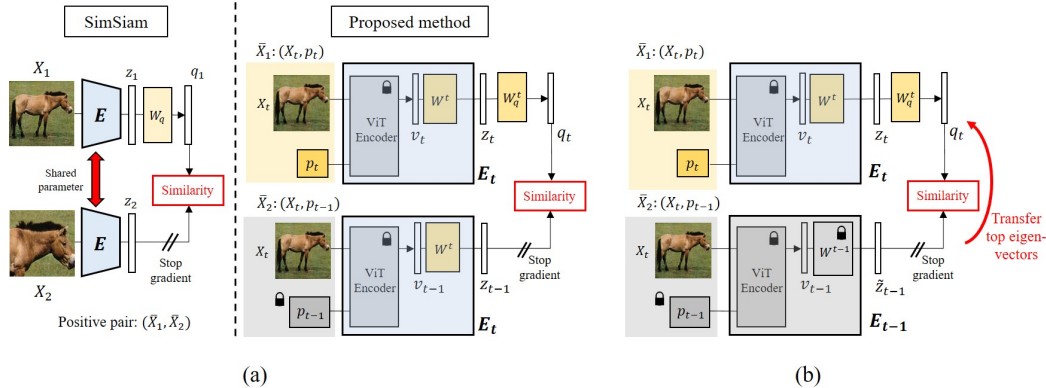

(a)                 (b)

Figure 4: Main concepts of the proposed method. (a) The prompt-augmented pairs are pulled in the embedding space to increase the dependency by the shared encoder $E_t$, similar to the Sim-Siam (Chen & He, 2021). (b) Transfer of top eigenvectors from the previous encoder $E_{t-1}$.

## 3.3 KNOWLEDGE ACCUMULATION BY PROMPT

**Information-theoretic view for continual learning**     We explore the method to accumulate the task knowledge by the prompt for continual learning. We suggest the information-theoretic view for the continual learning process, to clarify the motivation.

For the pretrained transformer $f$ with input $X_t$ and label $Y_t$ for the current task $t$, we denote $V_t = f(X_t, p_t)$ as the feature by current prompt $p_t$, and the $V_{t-1} = f(X_t, p_{t-1})$ as the feature by the previous prompt $p_{t-1}$. Then, we have the Markov chain $V_{t-1} \leftarrow X_t \rightarrow V_t \rightarrow Y_t$, which can be visualize as Figure 3(a). Then, we optimize the prompt by:

$$\max_{p_t} I(Y_t; V_t) + I(V_t; V_{t-1}) \tag{2}$$
$$= \max_{p_t} H(Y_t) - H(Y_t|V_t) + I(V_t; V_{t-1})$$

where $H(Y_t)$ is constant and the $H(Y_t|V_t)$ is minimized by the classification loss for current task. Now we focus on the $I(V_t; V_{t-1})$. We propose to use the positive-only contrastive loss, regarding $\bar{X}_1 : (X_t, p_t)$ and $\bar{X}_2 : (X_t, p_{t-1})$ as the positive pair $(\bar{X}_1, \bar{X}_2)$, as illustrated in Figure 3(b). We pull the pairs $(V_t, V_{t-1})$ in the embedding space, to increase the dependency between the variables. We reveal that the pulling the pairs enhances the alignment of the feature space, which leads to the preservation of the previous knowledge.

**SimSiam for the prompt-augmented pair** To enhance the interdependence between the prompt-augmented pair $(V_t, V_{t-1})$, we adopt the contrastive loss which pulls the pairs in the embedding space. Our approach is motivated by the resemblance between the recent positive-only contrastive methods and the framework of continual learning, as shown in Fig. 4.

Specifically, we employ the loss function of SimSiam (Chen & He, 2021) with simple linear layers $W^t, W_q^t$ for the task $t$, as follows:

$$\min_{p_t} L_{ctr} := -\mathbb{E}_{x \sim P(X_t)} \left[ q_t^\top z_{t-1} \right] \tag{3}$$

where $P(X_t)$ is the distribution for the task $t$ image, and $q_t, z_{t-1}$ are $l_2$ normalized vectors. The positive-only methods such as SimSiam captures the invariant characteristics of the positive pairs, while discarding the features that shows higher variance than the certain threshold (Tian et al., 2021; Wang et al., 2021). Likewise, the layer $W^t$ captures the embedding space that represents the common features between the positive pairs formed by the different prompts. Considering the observation in Fig. 2, the prompts for different tasks tends to have similar top eigenvectors, whereas the similarity rapidly decreases for the remaining eigenvectors. Therefore, $W^t$ captures the shared subspace across the tasks.

Additionally, the prompt $p_t$ is optimized to generate an embedding space aligned with the space by the previous prompt $p_{t-1}$. It guides the prompt $p_t$ to expand the shared subspace with the previous task, thereby enhancing the preservation of the previous knowledge.

**Knowledge transfer through embedding space**

We transfer the knowledge through the embedding space generated by the previous encoder $E_{t-1}$, as shown in Fig. 4. Similar to the $L_{ctr}$, we pull the feature $q_t$ and the $\tilde{z}_{t-1}$ to increase the dependency between the pair $(v_t, v_{t-1})$. Specifically, we optimize the loss function as follows:

$$\min_{p_t} L_{prev} = -\mathbb{E}_{x \sim P(X)} \left[ q_t^\top \tilde{z}_{t-1} \right] \tag{4}$$

where

$$\tilde{z}_{t-1} = E_{t-1}(x, p_{t-1}) \tag{5}$$
$$= W^{t-1} \circ f(x, p_{t-1}).$$

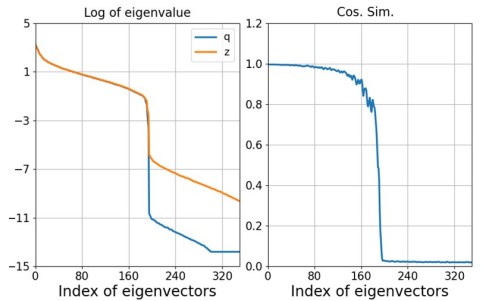

Figure 5: Eigenvalue and the eigenspace alignment for the $z_t$, $q_t$ feature spaces.

The authors in (Wang et al., 2021; Zhuo et al., 2023) reveals that the $q_t$ and $z_t$ has aligned eigenspace with reduced rank for $q_t$. As shown in Fig. 5, we present the empirical evidence of the property satisfied also in our framework, following Zhuo et al. (2023). Then, $L_{prev}$ can be seen to enhances the transfer of the top eigenvectors. It is similar to GPM (Saha et al., 2021) which utilize the top eigenvectors of the previous feature space to alleviate the forgetting. However, our method differs from GPM in that we imposes the constraint instead of using the memory buffer to collect the vectors.

### 3.4 OVERALL LOSS FUNCTION

Our method is to accumulate the knowledge in the prompt to alleviate the forgetting problem of the continual learning. We use the single prompt, hence, the proposed method do not require the task identification, which is essentially required for both the L2P and DualPrompt.

The overall diagram for the proposed method is shown in the Fig. 6(a). We employ the positive-only contrastive method (i.e. SimSiam (Chen & He, 2021)) as shown in Fig. 6(b). We impose the constraint on the prompt $p_t$ to pull the feature $V_t = f(X_t, p_t)$ and the previous feature $V_{t-1} = f(X_t, p_{t-1})$ in the embedding space, to accumulate the knowledge for continual learning. Specifically, the overall loss function is given as:

$$\min_{p_t} L = L_{cls} + \lambda_{ctr} L_{ctr} + \lambda_{prev} L_{prev} \tag{6}$$

where $\lambda_{ctr}, \lambda_{prev}$ are the hyperparameters. $L_{cls}$ is the cross entropy loss for the current classification task $\mathcal{D}_t$. We clarify our method by providing the pseudo code in the Appendix D.

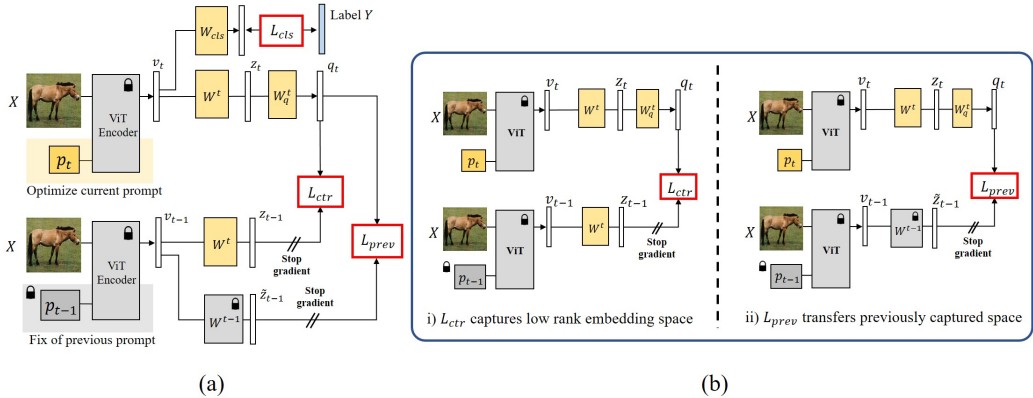

Figure 6: The proposed method is visualized in the following diagrams, with yellow colored blocks indicating the learnable components. (a) The overall framework is shown. (b) Two contrastive losses are jointly optimized to accumulate knowledge through prompts for continual learning.

## 4 METHODS

**Experimental settings** We verified our method with the Split-CIFAR100, Split-ImageNet-R datasets, Gaussian-scheduled CIFAR100, following the conventional prompt-based methods (Wang et al., 2022c;b). Split-CIFAR100 and Split-ImageNet-R datasets are for the class incremental learning (CIL) setting, where the tasks have disjoint classes. Gaussian-scheduled CIFAR100 is the task-agnositc continual setting, where the class shifts gradually during the training. It allows the overlapping classes between the tasks. We repeated the experiments 3 times with different seeds. More experimental details are summarized in the Appendix B.

**Weighted Projection Angle** In the Section 3.2, we investigate the forgetting of the eigenvectors by visualizing the *corresponding angle* (Zhu et al., 2021), where the value is averaged across the tasks. This metric significantly fluctuates unless it is averaged, since it measures the similarity between the vectors with the same index. Therefore, we propose the new metric, *weighted projection angle*, to effectively investigate the relation between the feature space of the current and the previous model, considering the eigenvalues of the previous space.

Specifically, we obtain the weighted projection matrix $M_o$ by old model $f_o$ using the current data $x_t \in \mathcal{D}_t$ as follows:

$$M_o = E_{x_t \in \mathcal{D}_t} \left[ f_o(x_t) f_o(x_t)^\top \right] = U \Lambda U^\top \tag{7}$$

where $U$ is the eigenvector matrix for the $d$-dimensional feature space $S_{old} = \text{span}\{u_1, \ldots, u_d\}$ of the previous model, and the $\Lambda$ is the eigenvalue matrix. $M_o$ projects the arbitrary vector $w$ to $\tilde{w} \in S_{old}$, allocating more weights for the direction of principal eigenvectors, as follows:

$$\tilde{w} = M_o w = \sum_{i=1}^{d} \lambda_i (u_i^\top w) v_i \tag{8}$$

Using the matrix $M_o$, we calculate projected vectors $\{\tilde{v}_i\}_{i=1}^{d}$ of eigenvectors $\{v_i\}_{i=1}^{d}$ for current model $f_t$ and current data $\mathcal{D}_t$. Then, the weighted projection angle is defined as:

$$cos(\phi_i) = \langle v_i, \tilde{v}_i \rangle / |\tilde{v}_i| \tag{9}$$

**Baseline methods** We compared our method with related continual learning methods. ER (Chaudhry et al., 2019b), BiC (Wu et al., 2019), DER++ (Buzzega et al., 2020), Co²L (Cha et al., 2021) are selected for rehearsal-based methods. The buffer size is set as 50 per class for Split-CIFAR100 and 25 per class for Split-ImageNet-R. EWC (Kirkpatrick et al., 2017), LwF (Li & Hoiem, 2017) are presented for the comparison with regularization methods. SupSup (Wortsman et al., 2020), DualNet (Pham et al., 2021), RPSNet (Rajasegaran et al., 2019) and DynaER (Yan

Table 1: Quantitative comparison of prompt-based continual learning methods. G-CIFAR100 refers the gaussian-scheduled CIFAR100 dataset.

| | Split-CIFAR100 (CIL) | | Split ImageNet-R (CIL) | | G-CIFAR100 (Task-agnostic) |
|---|---|---|---|---|---|
| | Average Accuracy↑ | Forgetting↓ | Average Accuracy↑ | Forgetting↓ | Average Accuracy↑ |
| Upper bound | 90.85 ±0.12 | - | 79.13 ±0.18 | - | 90.85 ± 0.12 |
| ER | 82.53 ±0.17 | 16.46 ±0.25 | 65.18 ±0.40 | 23.31 ±0.89 | 83.86 ± 0.38 |
| BiC | 81.42 ±0.85 | 17.31 ±1.02 | 64.63 ±1.27 | 22.25 ±1.73 | - |
| DER++ | 83.94 ±0.34 | 14.55 ±0.73 | 66.73 ±0.87 | 20.67 ±1.24 | 85.57 ± 0.41 |
| Co$^2$L | 82.49 ±0.89 | 17.48 ±1.80 | 65.90 ±0.14 | 23.36 ±0.71 | - |
| FT-seq | 33.61 ±0.85 | 86.87 ±0.20 | 11.14 ±0.58 | 79.91 ±0.64 | 38.94 ± 0.67 |
| EWC | 47.01 ±0.29 | 33.27 ±1.17 | 31.40 ±0.43 | 54.47 ±0.50 | 50.78 ± 0.60 |
| LwF | 60.69 ±0.63 | 27.77 ±2.17 | 14.93 ±0.13 | 63.29 ±0.23 | - |
| L2P | 84.50 ± 0.35 | 6.10 ± 0.22 | 62.49 ±0.49 | 11.45 ±0.51 | 84.17 ± 0.18 |
| DualPrompt | 85.18 ±0.49 | 5.48 ±0.25 | 69.44 ±0.31 | 5.23 ±0.13 | - |
| Proposed | **85.47 ±0.24** | **4.16 ±0.16** | **69.98 ±0.35** | **4.24 ±0.25** | **86.00 ± 0.17** |

Table 3: Quantitative comparison for varying the task number $T$. We present the results for 5 tasks (40 classes per task), 10 tasks (20 classes per task) and 20 tasks (5 classes per task).

| | 5 Tasks | | 10 Tasks | | 20 Tasks | |
|---|---|---|---|---|---|---|
| | Average Accuracy↑ | Forgetting↓ | Average Accuracy↑ | Forgetting↓ | Average Accuracy↑ | Forgetting↓ |
| FT-seq | 19.01 ±0.85 | 75.92 ±0.08 | 11.14 ±0.58 | 79.91 ±0.64 | 6.73 ± 0.42 | 84.81 ± 0.92 |
| LwF | 25.83 ± 0.29 | 50.53 ± 0.46 | 14.93 ± 0.13 | 63.29 ± 0.23 | 7.52 ± 0.03 | 73.90 ± 0.27 |
| L2P | 65.50 ± 0.59 | 6.97 ±0.58 | 62.49 ±0.49 | 11.45 ±0.51 | 58.49 ± 0.49 | 16.28 ± 0.78 |
| DualPrompt | 71.09 ±0.53 | 4.25 ±0.07 | 69.44 ±0.31 | 5.23 ±0.13 | 68.03 ± 0.26 | 6.42 ± 0.31 |
| :Oracle prompt | 71.70 ±0.23 | 3.71 ±0.21 | 70.82 ±0.15 | 4.76 ±0.21 | 71.12 ± 0.30 | 5.02 ± 0.29 |
| (Task ID Accuracy) | (56.50 ± 0.35) | | (45.89 ± 0.23) | | (40.83 ± 0.42) | |
| Proposed | **71.11 ±0.22** | **3.74 ±0.17** | **69.98 ±0.35** | **4.24 ±0.25** | **68.13 ± 0.32** | **5.32 ±0.33** |

et al., 2021) are selected for the previous architecture-based approaches. We also provide the result for naive training by prompt (FT-seq) and the supervised learning with all data (Upper bound). Lastly, we compared our method with prompt-based method, which are L2P (Wang et al., 2022c) and DualPrompt (Wang et al., 2022b). For G-CIFAR100, the results for BiC (Wu et al., 2019), Co$^2$L (Cha et al., 2021), LwF (Li & Hoiem, 2017) and DualPrompt (Wang et al., 2022b) are not presented, since the methods require the explicit task boundary, whereas the dataset provides the smooth transition of the tasks.

## 5 EXPERIMENTAL RESULTS

**Results** In Table 1, we present the results for CIL datasets which provide the explicit task boundary, as there is no overlap classes between the tasks. The results verifies the outperformance of our method compared to the previous methods. Specifically, it is remarkable that our method outputs better results without the memory buffer which is required for the rehearsal methods. Moreover, our method shows better result only with the single prompt, compared to the previous prompt-based methods with multiple prompts, such as L2P and DualPrompt. We additionally compared our method with architecture-based methods, as shown in Table 2. Our method outperforms the previous methods and the other backbones. Especially, our method tightens the accuracy gap without the memory buffer, which has advantage of being free from the data-privacy issue.

Table 2: Result for Split-CIFAR100 data by architecture-based approaches.

| Method | Backbone | Buffer Size | Avg. Acc.↑ | Acc. gap↓ |
|---|---|---|---|---|
| Upper-bound | | - | 80.41 | - |
| SupSup | | 0 | 28.34 | 52.07 |
| DualNet | ResNet18 | 1000 | 40.14 | 40.27 |
| RPSNet | | 2000 | 68.60 | 11.81 |
| DynaER | | 2000 | 74.64 | 5.77 |
| Upper-bound | ResNet152 | - | 88.54 | - |
| DynaER | | 2000 | 71.01 | 17.53 |
| Upper-bound | | - | 90.85 | - |
| L2P | ViT-B/16 | 0 | 83.86 | 6.99 |
| DualPrompt | | 0 | 85.18 | 5.67 |
| **Proposed** | | **0** | **85.47** | **5.38** |

Task-agnostic dataset (e.g. G-CIFAR100) presents the gradual transition of classes, where the task boundary cannot be defined. Therefore, some previous works (BiC, Co$^2$L, LwF, DualPrompt) are not applicable, as the methods require the explicit task boundary. In contrast, our method maintains its effectiveness also for the smooth transition of tasks, thanks to the accumulation of the knowledge in the prompt. As presented in Table 1, our method outperforms the other methods.

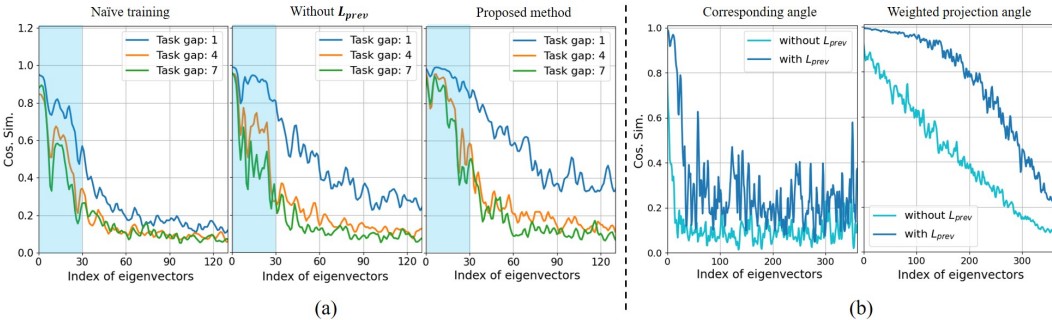

Figure 7: An analysis of the feature spaces is presented with the help of the following diagrams. (a) The similarity of the eigenvectors for $V_t$ feature spaces between tasks is visualized, with colored regions for comparison. (b) The corresponding angle and the weighted projection angle on the previous space for $Z_t$ are shown. The transfer of top eigenvectors is enhanced by $L_{prev}$.

**Limitation of prompt pool** We provide the additional results to clarify the limitation of the previous approach based on the prompt pool. We argue that the prompt pool has limitation due to its fixed number of prompt, as the pool size is predefined in advance. Moreover, we claim the degradation of performance by the misaligned prompt due to the inaccurate prediction of task identity in the inference stage, which is exacerbated as the task number increases.

To verify the claim, we fixed the size of the prompt pool of the L2P, and observed how the performance changes with the increase of the task number. For DualPrompt, we calculate the accuracy for task prediction, and reveal the performance gap by the misprediction. Split ImageNet-R dataset is used. As shown in Table 3, L2P shows the degradation as the task number increases. Likewise, the task prediction is getting lowered for DualPrompt as claimed. We also present the result for DualPrompt with ideal prompt selection for the given image (i.e. 'Oracle prompt' in Table 3). The task gap is increased as the accuracy for task prediction is degraded. Compared to the previous method, we observed the superiority of our approach, accumulating the knowledge in the prompt for the continual learning.

Furthermore, Table 4 compares the computational cost between our method and the previous methods. First, the proposed method requires far less parameters for the prompt to instruct the model, as our approach accumulate the knowledge in the single prompt. However, the previous approach captures the previous knowledge using a set of prompts, which introduces the redundancy in the parameters. Second, our method shows faster inference speed compared to the previous approach, thanks to the elimination of the prompt selection process. Specifically, our method simply employs the final prompt, whereas the previous methods are required to get features from the large frozen model to select the prompt from the prompt pool. We provide the details for the calculation of the computational costs in Appendix C.

Table 4: Comparison for the computational costs

|  | # Prompt Params. | Inference time (s) |
| --- | --- | --- |
| L2P | 0.92M | 41.61 |
| DualPrompt | 0.94M | 32.63 |
| **Ours** | **0.11M** | **25.04** |

**Analysis on the feature space** We analyze the change of the feature space for $V_t, Z_t$ during the continual learning, and verify the effectiveness of the proposed accumulative prompt to alleviate the forgetting problem. We visualize the corresponding angle (Zhu et al., 2021) of the eigenvectors, similar to the Section 3.2. The results in Figure 7(a) shows the enhanced similarity of the top eigenvectors by the proposed method. Specifically, as shown in the second graph, $L_{ctr}$ alone shows the limited improvement for the forgetting problem, although it contributes to the alignment between the intermediate tasks (i.e. Task gap 1 in the middle graph). The proposed method, which utilizes both $L_{ctr}$ and $L_{prev}$, largely increased the similarity also for more task gaps.

Similarly, Figure 7(b) supports the alignment of $Z_t$ space. We calculate the angles between the feature spaces of the final model $f_T$ and the former model $f_{T-1}$. The corresponding angle is enhanced at the front part of the index by the $L_{prev}$, which verifies the transfer of the top eigenvectors in the embedding space. Since the corresponding angle shows the large fluctuation, we additionally

present the proposed metric, weigthed projection angle, to clarify the improvement. The weighted projection angle clearly shows the high similarity by the $L_{prev}$ with less fluctuation. This result refers both the enhanced alignment of the space and the stability of the proposed metric. The results for both $V_t$ and $Z_t$ spaces verifies that the proposed method accumulative prompt keeps the alignment of the feature space across the tasks, which result the alleviation the forgetting problem.

## 6 DISCUSSIONS

**Comparison with the upper-bound prompt** $p^*$
We present the comparison for the feature space of $V_t$ between the prompt-based methods and the multi-task model (i.e. the upper-bound model supervised by i.i.d sampled data for all tasks). We utilized the validation dataset of the first task. The dataset for the comparison is Split ImageNet-R with 10 sequential tasks. For the wighted projection angle, we project the eigenvectors of prompt-based method to the space of the multi-task model.

As shown in Fig. 8, the corresponding angles are increased for more indices of the eigenvector. Moreover, the weighted projection angle clearly shows the enhanced alignment with less fluctuation. The result supports our overall claim that the proposed accumulative prompt can be a key to access the upper-bound prompt $p^*$ for the sequence of the tasks.

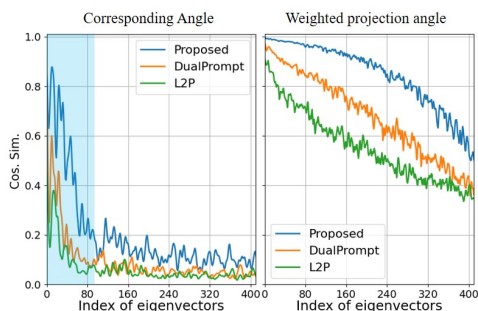

Figure 8: Feature space alignment with the upper-bound prompt. Our method shows the best alignment.

**Plasticity-Stability tradeoff** For continual learning, the model should be stable for previous task, while being adaptable for the new tasks. However, there exists a tradeoff between the stability and the adaptability (Mirzadeh et al., 2020; Wang et al., 2022a). We present the ablation study on the $\lambda_{prev}$, to analyze the tradeoff in our method.

In Fig. 9, we visualize the alignment of the feature space with varying $\lambda_{prev}$, using Split-ImageNet-R dataset. Averaged values for task gap 1 are presented. As $\lambda_{prev}$ increases, the cosine similarity between the eigenvectors also increases. The result demonstrates the improved stability of the model for the previous tasks, by the enhanced alignment between the feature spaces.

The result in Table 5 also verifies the improved stability by decreased forgetting. However the result also shows the lowered accuracy by the reduced plasticity of the model, which presents the plasticity-stability tradeoff. Still, our method has better tradeoff compared to the previous methods, showing higher accuracy than the other methods in Table 1.

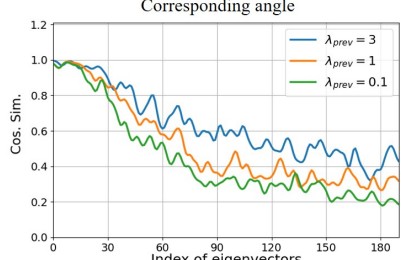

Figure 9: Alignment of feature space for $V_t$ with varying $L_{prev}$.

Table 5: Results for varying $\lambda_{prev}$, showing the trade-off.

| $\lambda_{prev}$ | Avg. Acc.↑ | Forgetting↓ |
|---|---|---|
| × | 69.59 | 5.35 |
| 0.1 | 69.85 | 4.88 |
| 2 | 69.71 | 3.93 |
| 3 | 69.56 | 3.77 |
| **Ours** | **69.90** | **4.10** |

## 7 CONCLUSION

We suggested a novel approach for prompt-based continual learning to alleviate the forgetting problem by the knowledge accumulation in the prompt. We first observed the misalignment of the feature space by the prompt leads to the catastrophic forgetting. Inspired from the recent success of the contrastive learning, we proposed a novel approach based on the obervation that the current model and previous model provides two different view of the given data, which can be regarded as the positive pair. Specifically, our method employed the positive-only contrastive learning, and experimental results verified that the method alleviates the forgetting by the transfer of the top eigenvectors.

## ETHICS STATEMENT

Training large-scaled models often requires massive computational cost, which can contribute to environmental concerns, including increased carbon emissions. Moreover, the forgetting of the previously acquired knowledge is a significant loss, considering the extensive computational resources consumed to train the model. In this context, the prompt-based approach for the continual learning is a promising solution by preventing the forgetting problem using only a few learnable parameters. The prompt learning offers the reduced computational burden. Moreover, the continual learning technique optimizes the prompt to preserve the previously obtained knowledge of the large pretrained model. Additionally, the use of prompt mitigates the data privacy issues. By eliminating the need for a memory buffer, prompt-based methods prevent the exposure of the previous data. We believe that our work would be a meaningful step towards the efficient utilization of large-scale models.

## REPRODUCIBILITY

We describe the experimental settings and details to reproduce our results in Appendix A, B. We also provide the pseudo code to clarify our method in Appendix D. We will release our code upon publication,

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

# Appendix

## A    SETTINGS FOR THE OBSERVATION

In the Section 3.1 of the main paper, we investigate the forgetting process by analyzing the corresponding angleZhu et al. (2021) between the feature spaces. For the tasks $t$ and $t'$, we define the task gap $\tau$ as the difference between the task indices(i.e. $\tau = |t' - t|$), and investigate how the angle changes as the task gap increases.

We extract the features by the models $f_t$, $f_{t'}$ which are trained until the tasks $t, t'$. The input is fixed as the validation dataset of the first task, $\mathcal{D}_1 = \{(x_{i,1}, y_{i,1})\}_{i=1}^{n_1}$. From the output features, we obtain the eigenvectors of the feature spaces $S_t = \text{span}\{u_1, ..., u_d\}$ and $S_{t'} = \text{span}\{v_1, ..., v_d\}$

The corresponding angle between the spaces is the cosine similarity between the eigenvectors with the same index, given as below:

$$cos(\phi_i; t, t') = \frac{\langle u_i, v_i \rangle}{\|u_i\|_2 \|v_i\|_2} \tag{10}$$

where $i$ is the index of the eigenvectors. However, the corresponding anlge shows significant fluctuation, since the minor change of the order of the vectors results large change in the value. Therefore, we present the averaged value for the given task gap $\tau$ as follows:

$$cos(\phi_i)_{avg,\tau} = \frac{1}{T - \tau} \sum_{t=1}^{T-\tau} cos(\phi_i; t, t+\tau) \tag{11}$$

where $T$ is the total number of the tasks.

We investigate the forgetting problem in the both of the CNN model and the vision transformer model. For CNN model, we used the ResNet18He et al. (2016) network, and trained it from the scratch by 10 tasks of Split-CIFAR100Krizhevsky et al. (2009) dataset. For the transformer model, we employed ViT-B/16Dosovitskiy et al. (2020) model and trained it using the 10 tasks of Split-ImageNet-RHendrycks et al. (2021) dataset.

## B    EXPERIMENTAL DETAILS

We summarize the hyperparameters in the Table. 6. For the experiments by the varying task number $T$ with Split ImageNet-R, the identical hyperparameters are used (i.e.$\lambda_{ctr} = 1, \lambda_{prev} = 1$).

We used ViT-B/16 (Dosovitskiy et al., 2020) model with pretrained on ImageNet (Deng et al., 2009), with simple linear layers for the projector $W^t \in \mathbb{R}^{786 \times 384}$ and the predictor $W_q^t \in \mathbb{R}^{384 \times 384}$. $W^t$ has the batch normalization layer at the end, following the Sim-SiamChen & He (2021). The prompt length and the layers to input

Table 6: Hyperparameters for each dataset.

|  | $\lambda_{ctr}$ | $\lambda_{prev}$ |
| --- | --- | --- |
| Split CIFAR100 | 0.5 | 0.5 |
| Split ImageNet-R | 1 | 1 |
| G-CIFAR100 | 1 | 1 |

the prompt is identical with the single prompt of the prompt pool in DualPrompt, for fair comparison with recent methods DualPrompt (Wang et al., 2022b) and L2P (Wang et al., 2022c). Our implementation is based on the official source code with Jax[1], and its pytorch version[2].

## C    CALCULATION OF COMPUTATIONAL COSTS

**Number of prompt parameters**    We calculate the number of the prompt parameters to clarify the advantage of our approach. We argue that the proposed method effectively instructs the model using fewer parameters, compared to the previous approach based on the prompt pool.

We introduce the calculation for the number of parameters for each methods. We clarify that the prompt size are identical to the single prompt in Dualprompt, for the fair comparison with the previous approach. We calculate the size of prompt for dataset of 10-task split-CIFAR100.

---

[1]https://github.com/google-research/l2p
[2]https://github.com/JH-LEE-KR/dualprompt-pytorch

First, our method input the prompts in 5 layers, with the length 5 for first two layers and length 20 for the rest of the layers. The prompts have 768 dimensions and separately defined for the query and key for the transformer layers. Therefore, the number of parameters for the prompt is calculated as:

$$\text{\# param. in total} = 2 * 5 * 768 * 2 + 3 * 20 * 768 * 2 = 107,520 \tag{12}$$

Second, DualPrompt has two different prompts, g-prompt and e-prompt. Prompts are input to 5 layers of the ViT. g-prompts are input to the first two layers, with the length of 5. e-prompts are input to the rest three layers, with the length of 20. Moreover, the prompt pool collects 10 e-prompts to memorize the previous knowledge. Similarly, the prompts have 768 embedding, and defined separately for the query and key in ViT layers. Then, the number of parameters for the prompt is calculated as:

$$\text{\# param. of g-prompts} = 2 * 5 * 768 * 2 = 15,360 \tag{13}$$
$$\text{\# param. of e-prompts} = 3 * 20 * 768 * 2 = 92,160 \tag{14}$$
$$\text{\# param. in total} = 15,360 + 10 * 92,160 = 936,960 \tag{15}$$

Lastly, L2P collects 30 prompts in the prompt pool, where the length of all the prompts are idential as 20. Then, the number of the parameters is calculated as:

$$\text{\# param. of single prompt} = 2 * 20 * 768 = 30,720 \tag{16}$$
$$\text{\# param. in total} = 30,720 * 30 = 921,600 \tag{17}$$

Our method utilizes significantly fewer parameters for the prompts, to guide the model for continual learning. The result demonstrates the efficiency of the proposed approach, accumulating knowledge by the single prompt.

**Inference time** We use the dataset for 10 tasks of split-ImageNet-R to estimate the inference time. We merged the validation datasets of all tasks. Then, we measure the time to generate the predictions for all data within the merged datasets.

## D   PSEUDO CODE

---
**Algorithm 1** Training procedure for the prompt

---
**Input:** Learnable $\{p, W, W_q, W_{cls}\}$ with fixed ViT
**Output:** $\{p, W_{cls}\}$

1: Initialize $\{\widetilde{p}, \widetilde{W}\}$ w/o gradient
2: **for** $t = 1, 2, \ldots T$ **do**
3:     **if** $t = 1$ **then**
4:         Set $\lambda_{ctr} = 0; \lambda_{prev} = 0$                          ▷ $\{L_{ctr}, L_{prev}\}$ are not used
5:         Train $\{p, W_{cls}\}$ by $L_{total}$
6:         Replace parameters $\widetilde{p} \leftarrow p$
7:     **else if** $t = 2$ **then**
8:         Set $\lambda_{prev} = 0$                          ▷ $\{L_{prev}\}$ is not used
9:         Train $\{p, W, W_q, W_{cls}\}$ by $L_{total}$
10:         Replace parameters $\widetilde{p} \leftarrow p; \widetilde{W} \leftarrow W$
11:     **else if** $t > 2$ **then**
12:         Train $\{p, W, W_q, W_{cls}\}$ by $L_{total}$
13:         Replace parameters $\widetilde{p} \leftarrow p; \widetilde{W} \leftarrow W$

---

We summarize the training stages in Algorithm 1. To clarify the process, we introduce the learnable modules $\{p, W, W_q, W_{cls}\}$ and the accumulative modules $\{\widetilde{p}, \widetilde{W}\}$, which corresponds to $\{p_t, W^t, W_q^t, W_{cls}\}$ and $\{p_{t-1}, W^{t-1}\}$. After the end of each task, the parameters of $\{\widetilde{p}, \widetilde{W}\}$ are replaced by the newly obtained $\{p, W\}$ to prepare the next task

In the algorithm, we made division for $t = 1$, $t = 2$, $t > 2$ for the training, as the loss function changes for each step. For the first task (i.e. $t$=1), $L_{ctr}$ is not applicable since the previous prompt

$\{p^{t-1}\}$ is not provided. Moreover, $L_{prev}$ is not used for the first and second task (i.e. $t$=1,2), as it requires the previous module $\{W^{t-1}\}$.

## E  ADDITIONAL ABLATION STUDY

**Ablation for related methods**  We conduct the ablation study for the related methods, specifically focusing on the contrastive learning and the knowledge distillation (KD) techniques. First, we compare our method with KD methods, including PODNet (Douillard et al., 2020), VID (Ahn et al., 2019) and AFC (Kang et al., 2022). Moreover, we conduct the comparison with other contrastive losses such as InfoNCE (Oord et al., 2018), Barlow Twins (Zbontar et al., 2021) and VICReg (Bardes et al., 2022). We used 10 tasks of split-CIFAR100 dataset. As shown in Table 7, the proposed method outperforms the related methods.

Table 7: Comparison for related works.

|         | Avg. Acc. | Forgetting |
|---------|-----------|------------|
| PODNet  | 81.53     | 6.66       |
| VID     | 84.20     | 7.10       |
| AFC     | 82.12     | 6.46       |
| InfoNCE | 85.48     | 4.85       |
| Barlow  | 81.09     | 6.76       |
| VICReg  | 82.53     | 5.71       |
| **Ours** | **85.47** | **4.16**   |

**Ablation on prompt length** $l_p$  We additionally provide the ablation study on the length of prompt. We set the prompt length to be identical with the recent SoTA, DualPrompt (Wang et al., 2022b), for fair comparison. The length of the prompts are 5 for the first two layers, and 20 for the rest of the layers. For the ablation study, we set all the prompts to have same length, and change the length. We used 10 tasks of split-ImageNet-R dataset. As shown in Table 8, the prompt length with 5 shows the improved forgetting on the previous task with the lowered accuracy which indicates the lowered plasticity of

Table 8: Ablation study for $l_p$.

| $l_p$    | Avg. Acc.$\uparrow$ | Forgetting$\downarrow$ |
|----------|---------------------|------------------------|
| 5        | 69.51               | 3.79                   |
| 20       | 69.67               | 4.41                   |
| 10       | 69.88               | 4.34                   |
| 40       | 69.31               | 4.23                   |
| **Proposed** | **69.90**       | **4.10**               |

the model. The performance is consistent with the increase of the prompt length, which demonstrates the robustness of the method to the prompt length.

