# OpenReview forum: "Knowledge Accumulating Contrastive Prompt for Continual Learning"
_ICLR.cc/2024/Conference — Submitted to ICLR 2024_

### Official Review · Reviewer_fnKJ · 2023-10-28

**Soundness:** 3 good
**Presentation:** 3 good
**Contribution:** 2 fair
**Rating:** 5
**Confidence:** 5

**Summary:**

This paper analyzed a critical issue of prompt-based approaches in continual learning, i.e., the errors in selecting an appropriate prompts from the prompt pool. To alleviate this issue, the authors proposed to accumulate knowledge in a single prompt through a contrastive learning strategy and transfer of top eigenvectors. Experimental results demonstrate that their method can achieve comparable performance as some representative baselines with smaller parameter cost.

**Strengths:**

1. This paper is well-written and easy to follow. The motivation is clear and well supported by empirical analysis.

2. The proposed method is reasonable. It’s good to seen that the capacity of a single prompt can be comparable to a prompt pool, especially in continual learning.

**Weaknesses:**

1. Despite the clear motivation, the proposed method only marginally outperforms the relatively earlier baseline in this direction, i.e., DualPrompt (ECCV22). There have been many more recent work based on constructing a prompt pool, e.g., CODA-Prompt (CVPR’23) and HiDe-Prompt (NeurIPS’23). I'm concerned about whether there is enough room for further development of the core insight of this paper, i.e., learning all tasks in a single prompt. Although I appreciate that the proposed method uses less parameters than L2P and DualPrompt, the improvement seems to be less significant because the prompts are light-weight.

2. In addition to the results of parameter cost and inference cost, I would encourage the authors to further compare their training cost, as the use of contrastive learning usually requires more computation in training phase.

3. I find a recent work [1] also discussed some similar issues, such as the misaligned prompts and the use of contrastive learning. While not required, I encourage the authors to make comparisons (at least conceptually) with this work.

[1] Hierarchical Decomposition of Prompt-Based Continual Learning: Rethinking Obscured Sub-optimality. NeurIPS 2023.

**Questions:**

Please refer to the Weakness.

Besides, I would suggest the authors to consider other fine-tuning techniques, such as adapter and LoRa. They usually have better capacity than prompts to fit downstream distributions, and might make the proposed method much stronger.

---

> ### Author Response · Authors · 2023-11-20
>
> >**Q1. The reviewer is concerned about the further improvement of the core insight of this paper, i.e., learning all tasks in a single prompt.**
>
> As the reviewer comments, we suggest the view that the single prompt has enough capacity to capture the knowledge of all tasks, which is verified by the upper-bound prompt $p^*$.
>
> According to the reviewer's concern, we highlight the accuracy gap between the upper-bound model and the continual learning methods, as shown in the Table 1 of main paper. We believe that the single prompt can further tighten the gap, as the existence of the optimal point within the parameter space is already provided. Moreover, the accumulation of the task knowledge improves the generalizability of the model, as suggested in [1].
>
> Moreover, our work is beneficial for task-agnostic setting, since using the single prompt allows the overlap between the tasks. In contrast, prompt pool stores the task-specific prompts, which requires the explicit task boundary.
>
> [1]Wang, Qin, et al. "Continual test-time domain adaptation." in CVPR 2022.
>
>
> >**Q2. Comparison for the training cost is required, as the method utilizes contrastive learning which may demand significant computation.**
>
> Following to the Reviewer's comment, we compare the training time per iteration for the validation sets of all tasks, as shown in Table 9.
>
> For the training, our method has the similar cost with the prompt selection methods. The previous methods computes the token feature from the seperate frozen model, which is used to train the prompt key. Similarly, we compute the features from the previous model which is frozen during the current task. Moreover, our method utilizes the positive-only contrastive loss, which avoids the additional computational burden by the negative pairs.
>
> **Table 9**. Comparison of training time for Split ImageNet-R.
>
>
> |    | Ours|DualPrompt|L2P|
> |----|----|----|----|
> |Training(s/iter.)|0.979|0.981|1.293|
>
> >**Q3. While not required, I encourage the authors to make comparisons (at least conceptually) with HiDe-Prompt.**
>
> Thanks to the reviewer's fruitful comment, we recognize the HiDe-prompt which is a very recent work(11 October, 2023 Arxiv and accepted for NeurIPS 2023) publicly released after the submission of our paper(28 September, 2023). We clarify that we could not recognize HiDe-prompt during our research.
>
> We summarize the differences between HiDe-prompt and our method, as follows:
> - HiDe-prompt is based on the prompt selection process and prompt pool.
> - Contrastive loss with negative pairs is suggested to remove the instance-wise and task-wise overlaps, which is different motivation with our method.
> - Additional process and buffer memory are required to obtain old-task statistics.
> - Not applicable for smooth transition of data distribution.
>
> We provide the detailed explanation for each element. First, HiDe-prompt suggest to improve the prompt selection process for the prompt pool, using the representative power of the uninstructed features. Second, the method proposes the contrastive learning to reduce the overlap between the task, which requires negative pairs. Positive pairs $(h, \mu_c)$ are presented, where $h$ is feature extracted by the prompt and $\mu_c$ is corresponding class prototype of the current task. Also, negative pairs $(h,h')$ and $ \lbrace \(h,\mu_{c,i} \) \rbrace_{i=1}^{t-1}$are defined to minimize the overlaps between the instances and tasks.
>
> Third, the method utilizes the old-task statistics, which should be calculated for each training stage. It requires additional process and the memory. Lastly, the method requires the explicit task boundary which is not provided for the continual learning with smooth transition of the data distribution(i.e. G-CIFAR100 dataset). In contrast, our method is applicable for the task-agnostic dataset, as shown in Table 1 in the main paper.

---

### Official Review · Reviewer_1SQd · 2023-10-31

**Soundness:** 3 good
**Presentation:** 3 good
**Contribution:** 2 fair
**Rating:** 6
**Confidence:** 3

**Summary:**

This paper proposed a new prompt-based method to address the catastrophic forgetting issue in continual learning. Instead of learning a group of prompts to capture the previous knowledge, the key idea of the proposed method is to learn the knowledge in a single prompt. The proposed method uses contrastive learning to pull the features of two different augmented views  in the embedding space. Experimental results demonstrate SOTA performance in continual learning.

**Strengths:**

1. This paper is overall well-structured and easy-to-follow.

2. The proposed single prompt learning based approach is effective in performance, memory saving and time-efficient.

3. The authors have done comprehensive analyses on different continual learning benchmarks, module ablation study, etc.

**Weaknesses:**

1. My major concern on this paper is that the performance improvement of the proposed method over the most relevant SOTA method (i.e., DualPrompt) is mirror on all benchmarks. This make it questionable why learning a single prompt is an optimal solution than learning a pool of prompts.

2. Compared to these prompt learning-based baseline methods (e.g., L2P, DualPrompt), what is the advantage of the proposed method on learning a single prompt in continual learning is not very well justified.

3. The formulation of $L_{ctr}$ and $L_{prev}$ are very similar except that different W are used. It is unclear to me which part plays a more important role in the method. It will be great more discussions and experiments on how $\lambda_{ctr}$ and $\lambda_{prev}$ affect the model performance on different benchmark are provided to help better understand these two losses.

**Questions:**

See "Weaknesses".

---

> ### Author Response · Authors · 2023-11-20
>
> >**Q1. Clarify the main advantage of single prompt compared to the prompt pool**
>
> We appreciate the constructive discussion suggested by the reviewer. According to the reviewer's concern, we highlight two main points of this paper as follows:
> - The proposed method allows the overlaps between the tasks.
> - We present the novel research direction for the prompt-based method, as the single prompt already has enough capacity.
>
> We provide the detailed explanation for each element. First, we argue that our method allows the overlap, which enables our work to be applicable for the task-agnostic setting. Specifically, for the G-CIFAR100 dataset, we introduce 200 different tasks to the model, as the data distribution is smoothly changed. Then, the prompt-pool based methods requires 200 different task-specific prompts. Moreover, the overlap between the tasks induces the redundancy of the prompt, which also arises the difficulty in prompt selection process. In contrast, our method is free from the limitations as the prompt pool and the prompt selection process are not required. To support the claim, we provide the result for G-CIFAR100 by dual prompt with 200 task-specific prompts.
>
> **Table7.** Comparison with task-specific prompts for task-agnostic setting (G-CIFAR100)
> |    | Ours | DualPrompt| L2P |
> |----|----|----|----|
> | Acc. | 86.00 | 79.37 | 84.17 |
> |\# Prompt Params|0.11M|18.4M|0.92M|
>
> Second, we suggest that the single prompt already has an enough capacity to instruct the model for all tasks, which is verified by the upper-bound prompt $p^*$. Hence, we claim that the primary concern is the optimization process, as the existence of the optimal point is provided. Accordingly, we propose the novel research direction of the prompt-based continual learning methods to approach the optimal point by the single prompt, which removes the need of the prompt pool and the selection process.
>
>
> >**Q2.More discussions and experiments for each loss term, $L_{ctr}$ and  $L_{prev}$.**
>
> As introduced in the main paper, $L_{ctr}$  captures the embedding space to pull the defined positive pairs, and $L_{prev}$ aligns the feature space between the old model and the current model.
> Specifically for $L_{prev}$, we kindly remind the plasticity-stability tradeoff suggested in the section 6 of the main paper. $L_{prev}$ enhances the alignment of the feature space, which contributes to the stability of the model for the continual learning.
>
> Following to the reviewer's comment, we present the additional ablation study for the loss term, as follows:
>
> **Table8.** Additional ablation study for loss terms
> | $\lambda_{ctr}$ | 1 | 1 | 1 | 1 | 2 | 2 | 2 | 0.5 | 0.5 | 0.5 |
> |--- |--- |--- |--- |--- | --- | --- | --- | --- | --- |---|
> | $\lambda_{prev}$ | 1 | X | 0.1 | 0.3 | X | 0.1 | 0.3 | X | 0.1 | 0.3 |
> | **Acc.**  | 69.98 | 69.59 | 69.85 | 69.56 | 69.20 | 69.28 | 68.45 | 69.97 | 69.80 | 69.44 |
> | **Forget.** | 4.24 | 5.35 | 4.88 | 3.77 | 4.11 | 3.97 | 3.96 | 5.34 | 5.46 | 4.30 |
>
>
> The result show that increasing $L_{prev}$ supports that the loss term alleviates the forgetting by enhancing the stability of the model. In case of $L_{ctr}$, the loss term contributes to the alleviated forgetting, however, increased $\lambda_{ctr}$ lowers the accuracy, suggesting degraded stability-plasticity tradeoff.

---

> > ### Comment · Reviewer_1SQd · 2023-11-23
> > **Thanks for your response**
> >
> > I thank the authors for their response. I feel some of my concerns have been addressed, but I still share the concern with other reviewers that the major focus of this paper should be the task-agnostic setup which the proposed method could benefit rather than the current task-specific setup (where the improvement over the previous SOTA is marginal). The experiment results shown in Q1 should be a good starting point for the revision. Overall, I acknowledge the contribution of this paper and keep my initial rating, but won't be upset if the paper is rejected.

---

### Official Review · Reviewer_ieXN · 2023-11-01

**Soundness:** 2 fair
**Presentation:** 3 good
**Contribution:** 3 good
**Rating:** 5
**Confidence:** 5

**Summary:**

The paper proposes a novel prompt-based approach to continually learn new tasks using just a single prompt. It accumulates both the previous and current task’s knowledge  in a single prompt  using contrastive learning without negative pairs, thereby removing the need for a pool of prompts and a corresponding task-id prediction mechanism  to select the prompt during inference (as in previous works).

**Strengths:**

1. The application of contrastive learning without negative pairs on prompt based continual learning seems novel.

2. The proposed approach helps in reduction of parameters and inference time without loss in performance.

3. Writing is clear and easy to understand.

**Weaknesses:**

[1]. The contrastive learning is novel but in compared to the recent work [1,5] paper does not shows the SOTA results. The recent prompting based baselines shows much better result but are missing in the paper.

[2]. The approach in [1]  seems to outperform the proposed approach. One justification can be the the approach requires two passes through the ViT during inference: one pass with the old prompt and another with the new prompt (referred to as ensemble in the paper). However, without the ensemble also, the approach seems to perform better as can be inferred from table 3 of their paper. Similarly, look the other work [4,5].

[3]. The prompting based model is mostly expansion based approach (where prompt is the expansion parameter) where these approach leverages over the strong pretrained model. In the absence of the pretrained model how the approach behaves? There are few expansion based recent work [2,3] that does not leverages the pretraiend model can author show the result compared to these approach.

[4] In the ablation (Table-4) the author has shown the model performance and prompt parameters growth which is good. The different parameter growth vs model performance is missing. How the model will behave if the prompt parameter are increased? If the next tasks are complex we require model prompt parameter to adapt the novel task.

[5] The paper mentions that the prompt selection  mechanism in l2p and dual-prompt can introduce mis-alignment of prompts. I am a bit curious as to how much mis-alignment does each of the approaches have?


[1] “A Unified Continual Learning Framework with General Parameter-Efficient Tuning, ICCV-2023”
[2] "Exemplar-Free Continual Transformer with Convolutions", ICCV-2023
[3] "Dytox: Transformers for continual learning with dynamic token expansion, CVPR-2022"
[4] "S-Prompts Learning with Pre-trained Transformers: An Occam's Razor for Domain Incremental Learning. NeurIPS-2022"
[5] "Coda-prompt: Continual decomposed attention-based prompting for rehearsal-free continual learning. CVPR-2023"

**Questions:**

Please refer to the weakness section.

---

> ### Author Response · Authors · 2023-11-20
>
> >**Comment 1. The reviewer suggested the detailed comparison with the recent related works, including S-Prompt and CODA-Prompt.**
>
> Thanks to the reviewer's fruitful comment, we recognize the HiDe-prompt which is a very recent work (11 October, 2023 Arxiv and accepted for NeurIPS 2023) publicly released after the submission of our paper (28 September, 2023). We clarify that we could not recognize HiDe-prompt during our research.
>
> In response to the reviewer’s comment, We provide the comparison with the related work, as follows:
>
> **Table 5.** Comparison with related works.
> |  | Ours | S-prompt | CODA-prompt |
> |---|----|---|---|
> |Acc. |85.47|80.13|82.72|
> |Forget.|4.16|9.19|10.26|
>
> >**Comment 2. How the performance changes if the parameters of the model are not frozen?**
>
> According to the reviewer's comment, we changed the parameters of the model to be learnable. Then, the result shows a severe forgetting. We present the task accuracy after the task 3, for the brevity of the response.
>
> **Table 6.** Task accuracy after training task 3 by unfrozen model.
> |  | Task1 | Task2 | Task3 | Average |
> |---|---|---|---|---|
> |**Acc.** |0.1|15.6|85.2|33.63|
>
>
> >**Comment 3. The reviewer commented the change of performance by the prompt length**
>
> The reviewer is kindly reminded that we have already provided the ablation study for prompt length, in the Table 8 in the supplementary material. We enlarged the prompt parameters by increasing the prompt length. The result shows the insensitivity of the performance to the length of the prompt.
>
> >**Comment 4. The reviewer commented the change of performance by the prompt length**
>
> We appreciate for the constructive discussion. The reviewer is kindly reminded that we have presented the effect of misalignment of the prompt in the Table 3 in the main paper. As DualPrompt predicts the task identity to select the task-specific prompt, we calculate the accuracy of the task prediction, denoted as `Task ID Accuracy'. The accuracy is below 60$\%$, and the accuracy degrades as the number of the tasks increases.
> In case of L2P, the prompt index is trained by the diversity regularization, without the ground-truth index. Accordingly, the accuracy for task prediction cannot be defined. Instead, we suggest the result of task-agnostic setting(e.g. G-CIFAR100) in Table 1 of the main paper. The lower performance of L2P compared to our method indirectly indicates the misaligned prompt problem of L2P.

---

### Official Review · Reviewer_5sqM · 2023-11-01

**Soundness:** 2 fair
**Presentation:** 3 good
**Contribution:** 1 poor
**Rating:** 3
**Confidence:** 4

**Summary:**

The core idea of this paper is that the upper-bound prompt is the prompt optimized by the merged dataset for all tasks. To approximate the upper-bound prompt, drawing inspiration from contrastive learning, the authors treat the input along with the current and previous prompts as two different augmented views (i.e., positive pairs). Then, the authors pull the features of these positive pairs in the embedding space together to accumulate knowledge. Experimental results demonstrate the performance increase of their method in continual learning.

**Strengths:**

1. The writing is clear.
2. This work applies the contrastive loss in the self-supervised learning to the class-incremental learning.

**Weaknesses:**

1. If using the same prompt for different sessions, the prompt is essentially a set of parameters that are constantly being updated. From this pespective, the loss proposed in this paper is very similar to the regularization loss in LwF. To be more specific, the loss used in this paper requires that the current prompt and previous prompts be similar, essentially demanding that the output of the new model and the old model be similar. However, the authors do not compare their methods to any classic regularization techniques.

2. The comparison in this paper is insufficient, as it does not compare their method to the CODA-Prompt [1]. Different from most existing prompt-based incremental learning methods, the authors use the same prompt for different sessions. To support the technique selection contrary to most existing methods, more comparison is essential.

3. The authors do not provide the performance of using the upper-bound prompt. It is insufficient to only prove that the prompt has been close to the upper-bound prompt. Whether is the performance of using the upper-bound prompt higher than the performance of state-of-the-art prompt-based incremental learning methods using task-specific prompts (including CODA-Prompt and HiDe-Prompt [2])?

4. This method restricts the model ability to learn new tasks, so I suspect that it may not work when there is a large gap between the pre-trained data and new-task data. It is essential to use other pre-trained models or conduct experiments on other datasets, e.g., using the model pretrained on ImageNet-1K with MoCo v3 [2,3]. For more datasets, the authors can refer to [4].

[1] CODA-Prompt: COntinual Decomposed Attention-based Prompting for Rehearsal-Free Continual Learning, CVPR 2023.

[2] Hierarchical Decomposition of Prompt-Based Continual Learning: Rethinking Obscured Sub-optimality, NeurIPS 2023.

[3] SLCA: Slow Learner with Classifier Alignment for Continual Learning on a Pre-trained Model, ICCV 2023.

[4] Revisiting Class-Incremental Learning with Pre-Trained Models: Generalizability and Adaptivity are All You Need.

In summary, my main concern is that the argument in this paper is not convincing. Theoretically, when different sessions use the same prompt, the prompt becomes a set of continuously updating model parameters. In incremental learning, it is challenging for the prompt to both avoid forgetting old tasks and adapt to new tasks. The reason that most existing works use task-specific prompts is to avoid this problem. This work is totally based on the premise that the same prompt is used for each session, but the author does not delve into why task-specific prompts should be abandoned. Experimentally, the method mentioned in this paper has not been compared to state-of-the-art methods using task-specific prompts, e.g., CODA-Prompt. It is also unknown whether the upper-bound prompt can bring the significant performance increase. Building on the two aspects, I still cannot believe that using the same prompt for all sessions can outperform using task-specific prompts. To justify this claim, more evidence may be required.

**Questions:**

1. Can the authors provide more evidences (e.g., analyses or experimental results) for supporting the use of the same prompt for different sessions rather than the use of task-specific prompts for different sessions?

2. The loss in this paper is essentially a regularization loss to prevent forgetting. Compared to the regularization loss in LwF, which one is better? Why is the regularization loss in this paper better? This is a very important problem.

3. Can the authors provide the comparison to CODA-Prompt? It will be better if the authors can provide the comparison between their method and HiDe-Prompt.

4. When using the upper-bound prompt, how does the model perform?

5. How is the model performance when using different pre-trained models or different datasets?

---

> ### Author Response · Authors · 2023-11-20
> **Official Comment by Authors [1/2]**
>
> >**Comment1. Provide more evidence for supporting the advantage of single prompt compared to task-specific prompt.**
>
> We appreciate for the constructive comments. We highlight that the task-specific prompts require the explicit task boundary, which is not provided for the task-agnostic setting. Specifically, G-CIFAR100 dataset introduces 200 different tasks, allowing the overlaps between tasks. According to reviewer's concern, we present the result of DualPrompt with 200 task-specific prompts.
>
> **Table1.** Comparison with task-specific prompts for task-agnostic setting (G-CIFAR100)
> |    | Ours | DualPrompt| L2P |
> |----|----|----|----|
> | Acc. | 86.00 | 79.37 | 84.17 |
> |\# Prompt Params|0.11M|18.4M|0.92M|
>
> As shown in the table, the methods with task-specific prompt require more parameters, which becomes redundant by the overlaps between the tasks.
>
> >**Comment2. The comparisons with recent related works, including CODA-Prompt are required.**
>
> In response to the reviewer's comment we present the result of the CODA-prompt by our implementation, as follows:
>
> **Table2.** Comparison with related approaches by 10-split-CIFAR100. ViT-B/16 model pre-trained on ImageNet-21k is used. We set learning rate and the scheduler identical to the main paper, for fair comparison.
>
> | Sup-21k   | Upper-bound | Ours |  CODA-Prompt |S-Prompt| HiDe-prompt |
> |----|----|----|---|---|---|
> | Acc. | 90.85 | 85.47 | 82.72|80.13|92.61|
> | Forget. |-|4.16|10.26|9.19|3.16|
>
> Thanks to the reviewer's fruitful comment, we recognize the HiDe-prompt which is a very recent work(11 October, 2023 Arxiv and accepted for NeurIPS 2023) publicly released after the submission of our paper(28 September, 2023). We clarify that we could not recognize HiDe-prompt during our research.
>
> We provide the conceptual comparison with HiDe-prompt, which is summarized as follows:
> - HiDe-prompt is based on the prompt selection process and prompt pool.
> - Contrastive loss with negative pairs is suggested to remove the instance-wise and task-wise overlaps, which is different motivation with our method.
> - Additional process and buffer memory are required to obtain old-task statistics.
> - Not applicable for smooth transition of data distribution.
>
> First, HiDe-prompt propose to improve the prompt selection process for the prompt pool, by the uninstructed features. Second, the method proposes the contrastive learning to reduce the task-wise overlap, which requires negative pairs. Positive pairs $(h, \mu_c)$ are presented, where $h$ is feature extracted by the prompt and $\mu_c$ is corresponding class prototype of the current task. Also, negative pairs $(h,h')$ and $ \lbrace \(h,\mu_{c,i} \) \rbrace_{i=1}^{t-1}$are defined to minimize the overlaps between the instances and tasks.
>
> Third, the method utilizes the old-task statistics, which should be calculated for each training stage. It requires additional process and the memory. Lastly, the method requires the explicit task boundary which is not provided for the continual learning with smooth transition of the data distribution(i.e. G-CIFAR100 dataset). In contrast, our method is applicable for the task-agnostic dataset, as shown in Table 1 in the main paper.
>
> >**Comment3. More discussion for previous regularization, including LwF is required**
>
> We appreciate for the comments. The reviewer is kindly reminded that we have present the comparison with the regularization methods, in Table 7 of supplementary material. We compare our method with the previous related methods, which are AFC, PODNet(POD-flat loss), and VID.
> According to the comment, we provide the result for KD loss which is similar loss with LwF.
>
> **Table3.** Result by Split-CIFAR100 using pretrained ViT by ImageNet-21k.
> | Sup-21k   | Upper-bound | Ours | KD |
> |----|----|----|----|
> | Acc. | 90.85 | 85.47 | 84.98|
> | Forget. |-|4.16|5.82|
>
> Our method regularize the model in the feature space, differently to the KD loss and the LwF. We compare the KD and LwF in response to the reviewer’s comment. KD loss regulate the current model’s logit by old model’s logit, which requires both the current prompt $p_t$ and the old prompt $p_{t-1}$. In constrast, LwF utilize the prompt $p_t$ and the memorized last linear layer $\theta_o$.
>
> The  Table 3 suggests that the regularization by old models helps to alleviate the forgetting, rather than the additional gradient by memorized $\theta_o$. Moreover, in our framework, we can easily get the old model by using the old prompt $p_{t-1}$. The results supports our approach, increasing the mutual information between the $p_t$ and $p_{t-1}$, to accumulate the knowledge.

---

> ### Author Response · Authors · 2023-11-20
> **Official Comment by Authors [2/2]**
>
> >**Comment4. Detailed explanation about the upper-bound prompt and its performance.**
>
> The reviewer is kindly reminded that we have already provided the performance of the upper-bound prompt in the Table 1 in the main paper. In section 3.1 of main paper, we specified the upper-bound prompt $p^*$ which is optimized by the merged dataset of all tasks (i.e. $\mathcal{D}_{all}=\mathcal{D}_1 \cup \dots \cup \mathcal{D}_T$). Hence, the $p^*$  instructs the model to perform well for all tasks. In Table 1 in the main paper, we present the accuracy by the $p^*$ denoted as `Upper bound'.
>
> >**Comment5. The reviewer commented to present more results using different pretrained models**
>
> We provide the ablation study for different pretrained models in Table 3. The result shows that our method consistently outperforms DualPrompt and L2P for all models. The comparison with CODA-prompt shows dependency for model architecture.
>
>
> **Table4.** Ablation study for pretrained model with related methods by 10-split-CIFAR100. We set learning rate and the scheduler identical to the main paper, for fair comparison
> |  |  | Upper-bound |Ours|CODA-prompt|DualPrompt|L2P|
> |--|--|--|--|--|--|--|
> |Sup-21k|Acc.|90.85|85.47|82.72|85.87|84.50|
> ||Forget.|-|4.16|10.26|5.48|6.10|
> |iBOT|Acc.|82.21|70.94|72.95|70.86|69.49|
> ||Forget.|-|9.74|14.84|10.50|10.07|
> |iBOT-21k|Acc.|84.05|73.76|74.87|69.67|64.52|
> ||Forget.|-|12.63|13.20|16.01|18.38|

---

> > ### Comment · Reviewer_5sqM · 2023-11-23
> > **Comment**
> >
> > Thank you for the comprehensive feedback provided.
> >
> > In a task-agnostic setting, using a single set of prompts may yield better results compared to allocating a distinct prompt set for each task. However, this deviates from the primary challenge faced in class-incremental learning. Task-agnostic incremental learning (online continual learning) and class-incremental learning face significantly different challenges. Therefore, the rebuttal from the reviewers hasn't convinced me that employing a single prompt set, as opposed to allocating distinct prompt sets for each task, could contribute to addressing the primary challenges in class-incremental learning.
> >
> > Then, it might be more appropriate for this article to focus on comparing methods within the task-agnostic setting (online continual learning), emphasizing the role of these methods in addressing the unique challenges faced in online continual learning. This shift in focus could potentially be more fitting. I encourage the authors to consider revising and resubmitting this paper for the next conference.

---

### Meta-Review · Area_Chair_rGSm · 2023-12-08

**Metareview:**

The paper addresses the problem of continual learning and proposes an approach to accumulate task knowledge using a single prompt. The reviewers raised concerns about minor novelty, insufficient comparisons with prior methods, and small gains over SOTA. The rebuttal included new experiments (including a comparison with CODA-Prompt). However, the response was not sufficient to convince the reviewers and the AC that employing a single prompt is an effective way to address class-incremental learning. While one reviewer recommends borderline accept, they did not champion the acceptance of the paper. Based on the above reasons, the AC recommends rejection and encourages the authors to follow the suggestions provided by the reviewers to improve the paper (e.g., focusing on the task-agnostic setting).

**Justification For Why Not Higher Score:**

The raised concerns by the reviewers regarding novelty, experimental validation, and justification for the proposed approach are legitimate.

**Justification For Why Not Lower Score:**

N/A

---

### Decision · Program_Chairs · 2024-01-16

Reject